# Improved Method for Cryopreservation of Embryogenic Callus of *Fraxinus mandshurica* Pupr. by Vitrification

**Xueqing Liu** [1,†], **Yingying Liu** [1,†], **Xiaoqian Yu** [1], **Iraida Nikolaevna Tretyakova** [2], **Alexander Mikhaylovich Nosov** [3,4], **Hailong Shen** [1,5] **and Ling Yang** [1,5,*]

1. State Key Laboratory of Tree Genetics and Breeding, School of Forestry, Northeast Forestry University, Harbin 150040, China
2. Laboratory of Forest Genetics and Breeding, Institute of Forest V.N. Sukachev, Siberian Branch of Russian Academy of Sciences, 660036 Krasnoyarsk, Russia
3. Laboratory of Cell Biology, Institute of Plant Physiology K.A. Timiryazev, Russian Academy of Sciences, 127276 Moscow, Russia
4. Department of Plant Physiology, Biological Faculty, Lomonosov Moscow State University, 119991 Moscow, Russia
5. State Forestry and Grassland Administration Engineering Technology Research Center of Korean Pine, Harbin 150040, China
* Correspondence: yangl-cf@nefu.edu.cn
† These authors contributed equally to this work.

**Abstract:** In order to simplify the experimental procedure and treatment procedure, we preserved the embryonic callus (EC) of *Fraxinus mandshurica* more efficiently. In this paper, we established a method for cryopreservation of EC of *F. mandshurica* by vitrification. EC was subcultured for 7–10 days (d). Vigorous EC with good growth conditions were selected, and cryopreservation was performed by vitrification. The best pre-culture method was to pre-culture EC on 0.5 mol·L$^{-1}$ sucrose medium for 3 d, load and culture in the liquid woody plant medium (WPM) supplemented with 2 mol·L$^{-1}$ glycerol and 0.4 mol·L$^{-1}$ sucrose for 60 min, then dehydrate in 2 mL of plant vitrification solution 2 (PVS2) (30% glycerol + 15% dimethyl sulfoxide (DMSO) + 15% ethylene glycol + 0.4 mol·L$^{-1}$ sucrose + liquid WPM). EC was rewarmed in a 40 °C water bath for 2 min after cooling in liquid nitrogen. The procedure for cryopreservation of *F. mandshurica* EC by the vitrification method established in this experiment is relatively reliable. The results from the present study provide a technical reference for improving the cryopreservation of *F. mandshurica* EC.

**Keywords:** *Fraxinus mandshurica*; embryogenic callus; cryopreservation; vitrification; regeneration

## 1. Introduction

Cryopreservation is the only safe and reliable method for the long-term preservation of germplasm resources [1–5]. Depending on the principle of dehydration, there are two main types of cryopreservation methods: one is the slow-cooling method, the other is the vitrification method. The slow-cooling method has been used for many years and has a complete technical system with relatively stable results. However, it requires expensive cooling devices such as a programmable coolers or continuous coolers for strict temperature control. The effect of slow-cooling on EC of *Fraxinus excelsior* L. was better than other cryopreservation methods [6]. Vitrification is the process whereby the material to be cryopreserved is pre-cultured, loaded and dehydrated in a high concentration of protective agent to form a non-crystalline vitrification as soon as possible during the rapid cooling process and kept safely in this state in liquid nitrogen at −196 °C [7,8]. Before it has time to produce ice crystals, the material to be preserved rapidly goes into a glassy state, thus achieving a long-term preservation result [9,10]. The experimental conditions required for the vitrification method are relatively simple and do not require expensive apparatus. However, the high concentration of cryoprotectants under vitrification treatment can have



a toxic effect on the material. Therefore, it is important to choose the right vitrification solution for the vitrification of each species for ultra-low temperature preservation [11,12].

*Fraxinus mandshurica* Rupr. is distributed in Northeast China, North China, Shaanxi, Gansu, Hubei and other places. It is one of the three most valuable hard broad-leaved species in the northeastern forest region. This species is made of excellent material and is used for high-grade furniture, tools and special construction, but the excellent resources of water willow are limited [13]. In previous studies, we have successfully induced somatic embryos [14] and EC [15] in *F. mandshurica* and have been able to preserve EC in petri dishes for three consecutive years by regular subculture (switching to fresh medium every month). However, in exceptional circumstances (e.g., COVID-19), when regular succession cultures cannot be entered into the laboratory in time, or when contamination is caused by improper manual operation and environmental conditions, callus in vitro is subject to genetic variation, embryogenic loss and cell death. This can result in the loss of a lot of experimental material. This is why we are working on the development of cryoprotectants techniques for *F. mandshurica*. We have recently reported on a new system for the slow-cooling of EC of *F. mandshurica* [16]. However, the lack of a programmable cooler to precisely control the temperature range and rate of cooling has led to some concerns about the stability of EC preservation results. Therefore, we believe it is necessary to continue to develop new, simple and inexpensive methods to provide a reference for efficient and stable conservation of valuable tree species (e.g., *F. mandshurica*) resources through improved techniques for cryopreservation of *F. mandshurica* EC.

To simplify experimental procedures and handling steps and to preserve embryonic guardian tissue effectively, in this paper, we establish a method for cryopreservation of EC of *F. mandshurica* by vitrification and carry out experiments on the differentiation of callus and plant regeneration after cryopreservation. An efficient and stable technique for the cryopreservation of *F. mandshurica* EC by vitrification is established. This technique provides a feasible method for perfecting the preservation of *F. mandshurica* germplasm resources.

## 2. Materials and Methods

### 2.1. Plant Materials

The method of obtaining EC of *F. mandshurica* referred to the method of Yu et al. [16]. Specific methods: Immature seeds were collected in early August from free-pollinated parent trees present on the campus of Northeast Forestry University, Harbin, Heilongjiang Province, China (126°37′55″ E, 45°43′16″ N). The cotyledons of sterile immature zygotic embryos were cultured in a woody plant medium (WPM) with 0.1 mg·L$^{-1}$ 6-Benzylaminopurine (6-BA) and 0.15 mg·L$^{-1}$ 2, 4-Dichlorophenoxyacetic Acid (2,4-D) to obtain EC and subcultured every 4 weeks. The EC of Z2 (W2) type of *F. mandshurica* selected by Yu et al. [16] for 7–10 d was selected as the material for cryopreservation by vitrification.

### 2.2. Experimental Method

#### 2.2.1. Cryopreservation of EC by Vitrification

All the conditions of vitrification cryopreservation were designed by a single factor experiment. The basic culture conditions were the same as the cryopreservation of EC of *Anemarrhena asphodeloides* Bunge by vitrification [17]. The specific methods were as follows:

(1) Sucrose concentration selection: 1.0 g of EC was inoculated on solid WPM with different concentrations of sucrose (0, 0.3, 0.4, 0.5, 0.6, 0.7 and 0.8 mol·L$^{-1}$). It was cultured in the dark at 25 °C for 1 d. The pre-cultured EC was added to a 1.8 mL cooling tube, along with the loading solution (2 mol·L$^{-1}$ glycerol + 0.4 mol·L$^{-1}$ sucrose + liquid WPM), and treated at room temperature for 40 min. Afterward, the loading solution was removed, and 2 mL of plant vitrification solution 2 (PVS2) was added (30% glycerol + 15% DMSO + 15% ethylene glycol + 0.4 mol·L$^{-1}$ sucrose + liquid WPM). It was dehydrated in the ice water mixture for 40 min, then added to the cooling tube, which was kept in liquid nitrogen immediately. It was then rewarmed for 2 min at 40 °C water bath after 2 h. After rapid removal of PVS2, it was washed 4 times with loading

solution in the horizontal flow clean bench, at intervals of 10–15 min. Finally, the EC was evenly dispersed on the filter paper, and the excess water was absorbed with a pipette and transferred to WPM for dark culture at 25 °C. After 24 h, the relative survival percentage of cells was calculated.

(2) Pre-culture time selection: The sucrose concentration of the pre-culture with the highest relative survival percentage was selected, and different pre-culture times (0, 1, 2, 3, 4 and 5 d) were screened at room temperature, wherein 0 d was the control group. After loading, dehydration, rewarming and washing for 24 h, for different pre-culture times, we compared their effects on the relative survival of cells after vitrification cryopreservation of *F. mandshurica* EC.

(3) Loading time selection: The loading time (30, 40, 50, 60, 70 and 80 min) was determined by the treatments with higher cell relative survival percentage in pre-cultured sucrose concentration and time. The relative survival percentage of cells was detected after dehydration, cooling, rewarming and washing for 24 h.

(4) Dehydration time selection: Vitrification dehydration time was screened based on the highest relative survival percentage in pre-culture and loading time treatment. A total of 2 mL of PVS2 (30% glycerol + 15% DMSO + 15% ethylene glycol + 0.4 mol·L$^{-1}$ sucrose + liquid WPM) was added to the EC mixture, and the dehydration times were 30, 40, 50, 60 and 70 min. We then compared the effects of different dehydration times on the cell relative survival percentage after cryopreservation of *F. mandshurica* EC after vitrification.

(5) Rewarming method selection: The different rewarming methods (25 °C room temperature, 40 °C water bath and running water washing) were determined by the treatment with the highest cell survival percentage with sucrose concentration, pre-culture time, loading time and dehydration time. We then compared the effects of different rewarming methods on the cell relative survival percentage after cryopreservation of *F. mandshurica* EC after vitrification.

(6) Rewarming time selection: Based on the abovementioned experiments, the rewarming time (1, 2, 3, 4 and 5 min) was determined. Then the loading solution was washed for 24 h, and the cell survival percentage was detected.

(7) Recovery culture of EC after resuscitation was based on the Yu et al. [15] recovery culture method for EC of *F. mandshurica*. Specific methods: After cryopreservation and resuscitation by vitrification, the EC of *F. mandshurica* was restored in WPM with 0.1 mg·L$^{-1}$ 6-BA and 0.15 mg·L$^{-1}$ 2,4-D. Subculture multiplication was carried out after 15–20 d. Differentiation culture was performed on $\frac{1}{2}$MS with 1.0 mg·L$^{-1}$ 6-BA medium. Then, it was mature cultured on $\frac{1}{2}$MS with 1.0 mg·L$^{-1}$ ABA medium. Germination and rooting culture was performed on $^1/_3$MS medium with 0.01 mg·L$^{-1}$ NAA medium, and the images were taken.

### 2.2.2. Determination of the Cell Relative Survival Percentage and Observation of Recovery Culture

For the fresh weight measurement and 2,3,5-Triphenyltetrazolium chloride (TTC) staining method of callus refer to Yu et al. [16]. The relative survival percentage of cells after cryopreservation is expressed as the ratio of absorbance of treatment and control [17]. The following formula was used:

$$\textit{Relative cell relative survival percentage } (\%) = \frac{OD \ value \ of \ cryopreservation \ treatment}{OD \ value \ of \ unprocessed} \times 100$$

Observation and record of recovery culture: According to the method of Liu et al. [15], the proliferation, differentiation and seedling emergence of *F. mandshurica* EC were observed and recorded every 30 d. The following formulae were used:

$$\textit{Callus proliferation coefficient} = \frac{weight \ of \ callus \ after \ proliferation}{weight \ of \ calls \ during \ inoculation}$$

$$\text{Callus differentiation percentage (\%)} = \frac{number\ of\ callus\ differentiated\ from\ somatic\ embryos}{number\ of\ callus\ inoculated} \times 100$$

$$\text{Regenerated plant percentage (\%)} = \frac{number\ of\ somatic\ embryos\ germinated\ into\ seedlings}{number\ of\ somatic\ embryos\ inoculated} \times 100$$

*2.3. Statistical Analysis*

Data were sorted by Microsoft Excel 2007, and one-way ANOVA was conducted by SPSS (2015). Graphs are drawn using sigmapplot software (2011). All data were the mean ± standard deviation of three replicates. Duncan's method was used to compare the significance between the data.

**3. Results**

*3.1. Effects of Sucrose Concentration on Fresh Weight and Cell Survival Percentage of EC*

Different sucrose concentrations showed significant effects on the fresh weight of *F. mandshurica* EC after cryopreservation and resuscitation by vitrification ($p < 0.05$) (Table 1). Compared with the fresh weight of EC cultured with sucrose on the 7th and 14th day of culture, it was found that the weight gain of EC was not significant. However, the fresh weight of EC cultured with sucrose increased significantly on the 21st day of culture. The fresh weight of EC cultured with the same sucrose concentration increased gradually with the prolongation of the culture period. On the 60th day, with the increase of concentration, the weight increased significantly at first and then decreased significantly. Therefore, after treatment with 0.5 mol·L$^{-1}$ sucrose solution, the weight of EC increased to 1.82 g, which was the highest. When the concentration of sucrose was 0 mol·L$^{-1}$, the fresh weight of EC was 1.04 g, and no significant difference was observed compared with that on day 0.

**Table 1.** Effect of sucrose concentration on EC fresh weight of *F. mandshurica* after cryopreservation by vitrification.

| Concentration (mol·L$^{-1}$) | Culture Time (d) | | | | | |
|---|---|---|---|---|---|---|
| | 0 | 7 | 14 | 21 | 30 | 60 |
| 0 | 0.91 ± 0.02 a | 0.93 ± 0.01 a | 0.94 ± 0.02 a | 1.00 ± 0.01 c | 1.02 ± 0.02 b | 1.04 ± 0.01 c |
| 0.3 | 0.90 ± 0.01 a | 0.93 ± 0.02 a | 0.94 ± 0.01 a | 1.15 ± 0.02 b | 1.24 ± 0.04 b | 1.52 ± 0.03 b |
| 0.4 | 0.93 ± 0.05 a | 0.96 ± 0.03 a | 0.98 ± 0.02 a | 1.16 ± 0.02 b | 1.26 ± 0.03 b | 1.56 ± 0.04 b |
| 0.5 | 0.92 ± 0.06 a | 0.95 ± 0.05 a | 0.97 ± 0.03 a | 1.31 ± 0.03 a | 1.54 ± 0.04 a | 1.82 ± 0.03 a |
| 0.6 | 0.93 ± 0.01 a | 0.95 ± 0.01 a | 0.98 ± 0.02 a | 1.15 ± 0.02 b | 1.25 ± 0.04 b | 1.53 ± 0.03 b |
| 0.7 | 0.93 ± 0.05 a | 0.95 ± 0.03 a | 0.96 ± 0.01 a | 1.13 ± 0.02 b | 1.23 ± 0.03 b | 1.54 ± 0.04 b |

Note: The data in the table represent the fresh weight (g) of EC. The data in the table are expressed as the mean ± SE; different lowercase letters in the same column represent significant differences ($p < 0.05$).

Sucrose treatment significantly affected the cell survival percentage of EC after cryopreservation by vitrification (Figure 1). The cell survival percentage without sucrose pre-culture (0 mol·L$^{-1}$) was 8.83%, which was the control group. The cell survival percentage of the sucrose pre-culture was higher than that of the control. With the increase of concentration, the cell survival percentage increased significantly at first and then decreased significantly. After pre-culture with 0.3 mol·L$^{-1}$ sucrose, the cell survival percentage was significantly higher than that of the control group. After sucrose culture with 0.5 mol·L$^{-1}$, the cell survival percentage was the highest (81.86%) ($p < 0.05$). With the increase of sucrose concentration to 0.8 mol·L$^{-1}$, the cell survival percentage decreased to 20.41%. Therefore, 0.5 mol·L$^{-1}$ sucrose pre-culture is the best choice for vitrification cryopreservation of *F. mandshurica* EC.

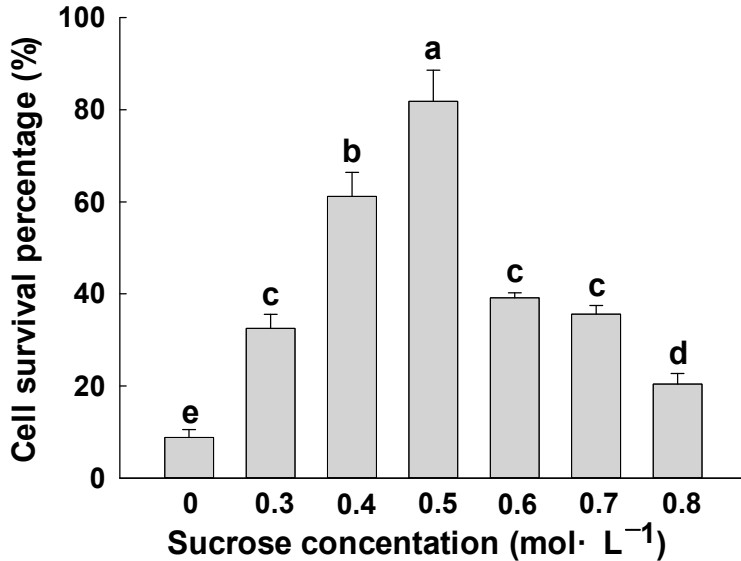

**Figure 1.** Effects of sucrose concentration on the cell survival of *F. mandshurica* EC after cryopreservation by vitrification. Note: Different letters are significantly different from each other at $p < 0.05$, using Duncan's Multiple Range Test.

*3.2. Effect of Pre-Culture Time on Fresh Weight and Cell Survival Percentage of EC*

The pre-culture time had a significant effect on the fresh weight of EC after vitrification cryopreservation (Table 2). On the 21st day of culture, compared with the day 0, the fresh weight of EC treated with different sucrose concentrations increased significantly. The fresh weight of each treatment increased significantly. In addition, on the same culture day, with the increase of pre-culture time, the weight increased significantly at first and then decreased significantly. When cultured for the 60th day, the fresh weight of the EC pre-cultured for 3 days was the highest (1.56 g). When the pre-culture time was 0 day, the fresh weight of EC reached the minimum value (1.13 g). This treatment is very different from other treatments ($p < 0.05$).

**Table 2.** Effect of pre-culture time on fresh weight of *F. mandshurica* EC after cryopreservation by vitrification.

| Pre-Culture Time (d) | Culture Time (d) | | | | | |
|---|---|---|---|---|---|---|
| | 0 | 7 | 14 | 21 | 30 | 60 |
| 0 | 0.93 ± 0.02 a | 1.00 ± 0.03 a | 1.02 ± 0.03 a | 1.09 ± 0.02 c | 1.11 ± 0.01 c | 1.13 ± 0.02 c |
| 1 | 0.92 ± 0.03 a | 0.99 ± 0.05 a | 1.01 ± 0.05 a | 1.14 ± 0.02 b | 1.27 ± 0.03 b | 1.35 ± 0.03 b |
| 2 | 0.93 ± 0.02 a | 1.01 ± 0.04 a | 1.03 ± 0.03 a | 1.18 ± 0.02 a | 1.35 ± 0.03 a | 1.37 ± 0.03 b |
| 3 | 0.92 ± 0.01 a | 1.00 ± 0.03 a | 1.03 ± 0.03 a | 1.19 ± 0.04 a | 1.36 ± 0.03 a | 1.56 ± 0.04 a |
| 4 | 0.94 ± 0.02 a | 1.00 ± 0.04 a | 1.02 ± 0.04 a | 1.14 ± 0.01 b | 1.15 ± 0.02 c | 1.35 ± 0.04 b |
| 5 | 0.93 ± 0.01 a | 0.99 ± 0.03 a | 1.01 ± 0.02 a | 1.14 ± 0.02 b | 1.26 ± 0.03 b | 1.34 ± 0.05 b |

Note: The data in the table represent the fresh weight (g) of EC. The data in the table are expressed as the mean ± SE; different lowercase letters in the same column represent significant differences ($p < 0.05$).

Pre-culture time showed a significant effect on the relative survival percentage of EC during vitrification cryopreservation (Figure 2). When the control group was pre-cultured for day 0, the cell survival percentage was 9.51%. When it was pre-cultured for 3 days, the cell relative survival percentage was the highest (66.64%). The difference between the two groups was significant ($p < 0.05$). With a gradual increase in pre-culture days, the cell relative survival percentage first increased and then decreased. When pre-cultured for 5 days, the cell survival percentage decreased to 28.28%. Thus, the optimal sucrose pre-culture time before cryopreservation of *F. mandshurica* EC was 3 days.

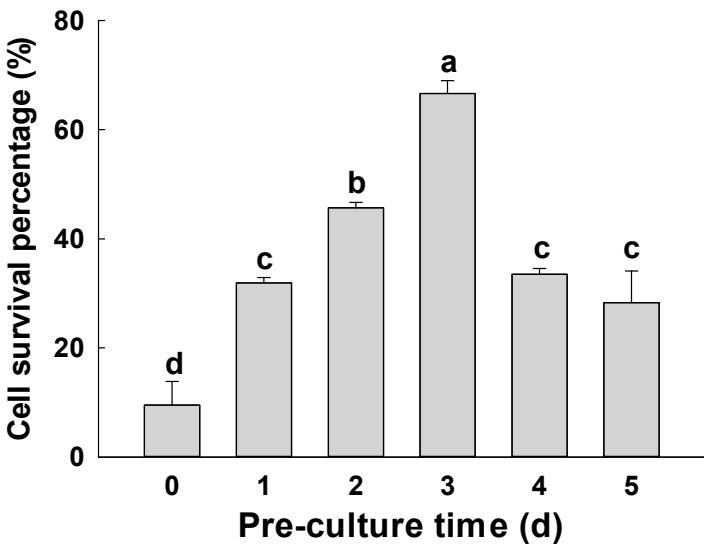

**Figure 2.** Effects of pre-culture time on cell survival percentage of *F. mandshurica* EC after cryopreservation by vitrification. Note: Different letters are significantly different from each other at $p < 0.05$, using Duncan's Multiple Range Test.

*3.3. Effect of Loading Time on Fresh Weight and Cell Survival Percentage of EC*

In the process of vitrification cryopreservation, different loading time had a significant effect on the fresh weight of EC (Table 3). On the same culture day, with the increase of loading time, the weight increased significantly at first and then decreased significantly. On the 60th day, the fresh weight of EC treated for 60 min was the highest (1.76 g), which was significantly different from that of other loading time ($p < 0.05$). When the loading time was 30 min, the fresh weight of EC was 1.11 g, which was the smallest.

**Table 3.** Effect of loading time on fresh weight of *F. mandshurica* EC after vitrification cryopreservation.

| Loading Time (min) | Culture Time (d) | | | | | |
|---|---|---|---|---|---|---|
| | 0 | 7 | 14 | 21 | 30 | 60 |
| 30 | 0.96 ± 0.02 a | 1.00 ± 0.02 a | 1.05 ± 0.01 a | 1.09 ± 0.02 c | 1.10 ± 0.02 d | 1.11 ± 0.01 e |
| 40 | 0.95 ± 0.03 a | 1.00 ± 0.02 a | 1.03 ± 0.01 a | 1.14 ± 0.03 b | 1.27 ± 0.03 bc | 1.34 ± 0.07 c |
| 50 | 0.95 ± 0.02 a | 0.99 ± 0.02 a | 1.02 ± 0.02 a | 1.24 ± 0.02 a | 1.31 ± 0.02 a | 1.44 ± 0.08 b |
| 60 | 0.97 ± 0.02 a | 1.01 ± 0.02 a | 1.03 ± 0.02 a | 1.27 ± 0.02 a | 1.37 ± 0.04 a | 1.76 ± 0.03 a |
| 70 | 0.96 ± 0.01 a | 1.01 ± 0.02 a | 1.04 ± 0.03 a | 1.12 ± 0.03 b | 1.21 ± 0.04 c | 1.51 ± 0.03 b |
| 80 | 0.96 ± 0.01 a | 0.98 ± 0.02 a | 1.03 ± 0.02 a | 1.08 ± 0.04 bc | 1.21 ± 0.03 c | 1.24 ± 0.03 d |

Note: The data in the table represent the fresh weight (g) of EC. The data in the table are expressed as the mean ± SE; different lowercase letters in the same column represent significant differences ($p < 0.05$).

Different loading time had a significant effect on the cell survival percentage of EC during vitrification cryopreservation (Figure 3). With the increase of loading time, the cell survival percentage increased significantly at first and then decreased significantly. When the loading time was 30 min, the cell survival percentage was the smallest (20.24%). When the loading time was 60 min, the cell survival percentage was the highest (61.56%) and significantly different from others ($p < 0.05$). When the loading time was 80 min, the cell survival percentage decreased significantly. Thus, the optimal loading time for cryopreservation of *F. mandshurica* EC by vitrification was 60 min.

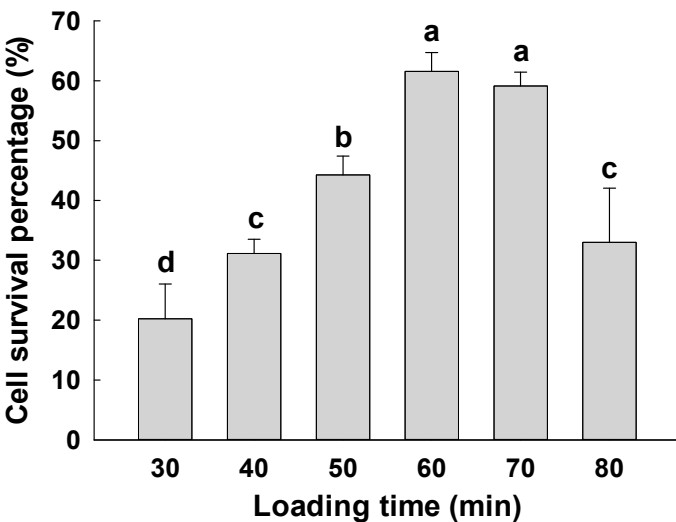

**Figure 3.** Effects of loading time on cell survival percentage of *F. mandshurica* EC after cryopreservation by vitrification. Note: Different letters are significantly different from each other at $p < 0.05$, using Duncan's Multiple Range Test.

*3.4. Effects of Dehydration Time on Fresh Weight and Cell Survival Percentage of EC*

In the process of vitrification cryopreservation, dehydration time showed a significant effect on the fresh weight of EC (Table 4). The fresh weight increased with the increase of recovery culture time. On the 60th day of culture, with the extension of dehydration time, the fresh weight of EC showed a trend of first increasing and then decreasing. When treated for 50 min, the maximum fresh weight of EC was 1.75 g, which was 80.41% higher than that of the resuscitation culture on day 0. When treated for 70 min, the lowest fresh weight of 1.29 g was observed, which was 31.63% higher than that of the resuscitation culture on day 0. A significant difference in the fresh weight of the callus was observed between 50 min and 70 min of dehydration time ($p < 0.05$).

**Table 4.** Effect of dehydration time on fresh weight of *F. mandshurica* EC after cryopreservation by vitrification.

| Dehydration Time (min) | Culture Time (d) | | | | | |
|---|---|---|---|---|---|---|
| | 0 | 7 | 14 | 21 | 30 | 60 |
| 30 | 0.98 ± 0.06 a | 1.01 ± 0.05 a | 1.04 ± 0.03 a | 1.12 ± 0.04 b | 1.22 ± 0.03 b | 1.34 ± 0.05 b |
| 40 | 0.97 ± 0.02 a | 0.99 ± 0.02 a | 1.01 ± 0.02 a | 1.16 ± 0.03 b | 1.32 ± 0.09 b | 1.69 ± 0.05 a |
| 50 | 0.99 ± 0.03 a | 1.01 ± 0.04 a | 1.02 ± 0.04 a | 1.27 ± 0.10 a | 1.48 ± 0.12 a | 1.75 ± 0.10 a |
| 60 | 0.97 ± 0.04 a | 1.02 ± 0.02 a | 1.03 ± 0.02 a | 1.11 ± 0.03 b | 1.19 ± 0.03 b | 1.38 ± 0.02 b |
| 70 | 0.98 ± 0.04 a | 1.00 ± 0.03 a | 1.02 ± 0.07 a | 1.14 ± 0.02 b | 1.21 ± 0.03 b | 1.29 ± 0.03 b |

Note: The data in the table represent the fresh weight (g) of EC. The data in the table are expressed as the mean ± SE; different lowercase letters in the same column represent significant differences ($p < 0.05$).

With different dehydration time, the cell survival percentage of EC after vitrification cryopreservation is also different (Figure 4). With the increase of dehydration time, the cell survival percentage of EC showed a trend of first increasing and then decreasing. When the dehydration time was 70 min, the cell survival percentage decreased sharply (29.13%). When treated for 50 min, the relative survival percentage was the highest (67.65%), which was significantly different from that of 70 min. Under a 30 min treatment, the cell survival percentage was 35.80%. Thus, the optimal dehydration time for the cryopreservation of *F. mandshurica* EC by vitrification was 50 min, and its relative survival percentage was also the highest.

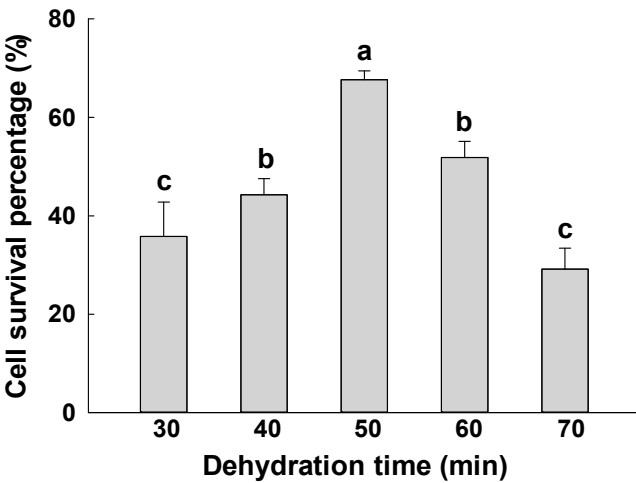

**Figure 4.** Effects of dehydration time on cell survival percentage of *F. mandshurica* EC after cryopreservation by vitrification. Note: Different letters are significantly different from each other at $p < 0.05$, using Duncan's Multiple Range Test.

### 3.5. Effects of Rewarming Methods on Fresh Weight and Cell Survival Percentage of EC

The method of rewarming affected the fresh weight of EC after vitrification cryopreservation, but the difference was not significant (Table 5). Under the rewarming treatment at 25 °C, the fresh weight of EC increased gradually with the increase of recovery culture time. On the 60th day of culture, the fresh weight of EC was the lowest (1.05 g), which was 8.25% higher than that on day 0. Under a 40 °C water bath treatment, the fresh weight of EC was the highest (1.33 g), which was 46.15% higher than that on day 0. For *F. mandshurica* EC after vitrification cryopreservation, the cell survival percentage was different with different rewarming methods (Figure 5). Under a 25 °C treatment, the cell survival percentage was 35.80%. Under a 40 °C water bath treatment, the cell survival percentage was the highest (61.84%). When the callus was washed with running water, the cell survival percentage was the lowest (22.71%). Thus, the 40 °C water bath is the most suitable rewarming method for the cryopreservation of *F. mandshurica* EC by vitrification.

**Table 5.** Effects of rewarming methods on the fresh weight of *F. mandshurica* EC after cryopreservation by vitrification.

| Rewarming Method | Culture Time (d) | | | | | |
|---|---|---|---|---|---|---|
| | 0 | 7 | 14 | 21 | 30 | 60 |
| 25 °C room temperature | 0.97 ± 0.03 a | 0.99 ± 0.02 a | 0.99 ± 0.01 a | 1.01 ± 0.01 c | 1.02 ± 0.01 c | 1.05 ± 0.02 c |
| 40 °C water bath | 0.91 ± 0.02 a | 0.94 ± 0.03 a | 0.96 ± 0.02 a | 1.14 ± 0.01 a | 1.19 ± 0.01 a | 1.33 ± 0.04 a |
| Running water | 0.92 ± 0.05 a | 0.94 ± 0.04 a | 0.98 ± 0.04 a | 1.05 ± 0.03 b | 1.12 ± 0.02 b | 1.18 ± 0.03 b |

Note: The data in the table represent the fresh weight (g) of EC. The data in the table are expressed as the mean ± SE; different lowercase letters in the same column represent significant differences ($p < 0.05$).

### 3.6. Effect of Rewarming Time on Fresh Weight and Cell Survival Percentage of EC

Under a 40 °C water bath treatment, the fresh weight of *F. mandshurica* EC was different with different rewarming times. With the extension of rewarming time, the fresh weight of EC showed a trend of first increasing and then decreasing (Table 6). The fresh weight of EC increased gradually with an increase in recovery culture time. When the rewarming time was 2 min, the fresh weight of EC cultured on the 60th day was the highest (1.47 g), which increased by 59.78% compared with the recovery culture on day 0. When treated for 1 min, the minimum fresh weight of EC was 1.17 g, which was 24.47% higher than that on day 0. Different rewarming times showed a significant effect on the relative survival percentage of EC of *F. mandshurica* after cryopreservation by vitrification (Figure 6). The cell survival

percentage first increased and then decreased with an increase in rewarming time. When treated for 2 min, the cell survival percentage was the highest (62.13%). However, when the rewarming times were 1 min and 5 min, the cell survival percentages were lower, namely 15.11% and 22.93%. When treated for 4 min, the cell survival percentage decreased sharply and was significantly different from that at 2 min ($p < 0.05$).

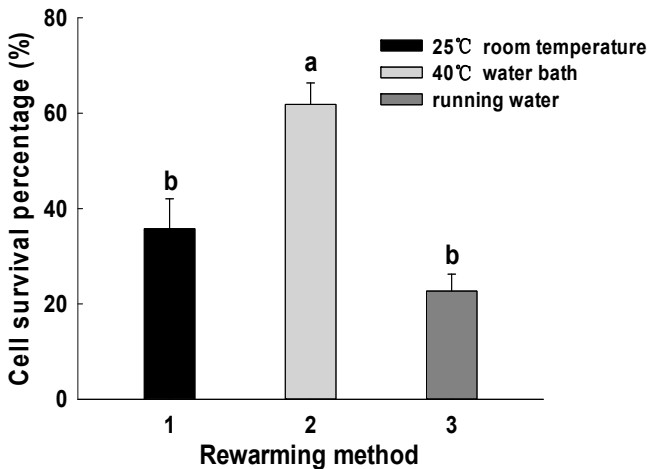

**Figure 5.** Effects of rewarming method on cell survival percentage of *F. mandshurica* EC after cryopreservation by vitrification. Note: Different letters are significantly different from each other at $p < 0.05$, using Duncan's Multiple Range Test.

**Table 6.** Effect of rewarming time on fresh weight of *F. mandshurica* EC after cryopreservation by vitrification.

| Rewarming Time (min) | Culture Time (d) | | | | | |
|---|---|---|---|---|---|---|
| | 0 | 7 | 14 | 21 | 30 | 60 |
| 1 | 0.94 ± 0.01 a | 0.98 ± 0.01 a | 0.99 ± 0.02 a | 1.09 ± 0.05 ab | 1.12 ± 0.06 b | 1.17 ± 0.03 b |
| 2 | 0.92 ± 0.04 a | 0.95 ± 0.05 a | 0.99 ± 0.04 a | 1.14 ± 0.06 a | 1.26 ± 0.05 a | 1.47 ± 0.12 a |
| 3 | 0.92 ± 0.02 a | 0.94 ± 0.02 a | 0.97 ± 0.06 a | 1.15 ± 0.09 a | 1.21 ± 0.06 a | 1.28 ± 0.03 b |
| 4 | 0.90 ± 0.03 a | 0.93 ± 0.03 a | 0.98 ± 0.05 a | 1.01 ± 0.04 b | 1.10 ± 0.02 b | 1.22 ± 0.03 b |
| 5 | 0.94 ± 0.02 a | 0.94 ± 0.02 a | 0.97 ± 0.04 a | 1.03 ± 0.05 b | 1.11 ± 0.03 b | 1.19 ± 0.03 b |

Note: The data in the table represent the fresh weight (g) of EC. The data in the table are expressed as the mean ± SE; different lowercase letters in the same column represent significant differences ($p < 0.05$).

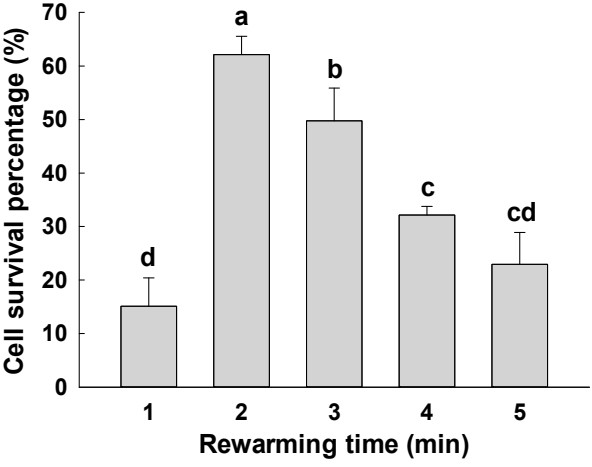

**Figure 6.** Effects of rewarming time on cell survival percentage of *F. mandshurica* EC after cryopreservation by vitrification. Note: Different letters are significantly different from each other at $p < 0.05$, using Duncan's Multiple Range Test.

### 3.7. Somatic Embryogenesis and Plant Regeneration after Resuscitation

The proliferation coefficient of the EC of *F. mandshurica* after cryopreservation by vitrification was 2.69, whereas the proliferation coefficient of uncryopreserved EC (control) was 2.86. When the callus was restored on day 60 of culture, the callus was loose and proliferated (Figure 7a). Thereafter, it was transferred to a differentiation medium. Although it could differentiate into somatic embryos normally, the number of somatic embryos was less (Figure 7b). The callus differentiation percentage was 53.87%, and the callus differentiation percentage of the control was 59.44%. After 2 months of germination and rooting culture, the somatic embryos germinated normally into seedlings (Figure 7c). The plant regeneration percentage of somatic embryos after cryopreservation was 20.97%, while that of somatic embryos without cryopreservation was 25.87%. A month later, somatic embryos were cultured in canned bottles for rooting (Figure 7d). To conclude, after cryopreservation of *F. mandshurica* EC by vitrification, the callus proliferation coefficient, callus differentiation percentage, and plant regeneration percentage decreased compared with the control, but the difference was not significant. This showed that the vitrification method is suitable for the preservation of *F. mandshurica* EC.

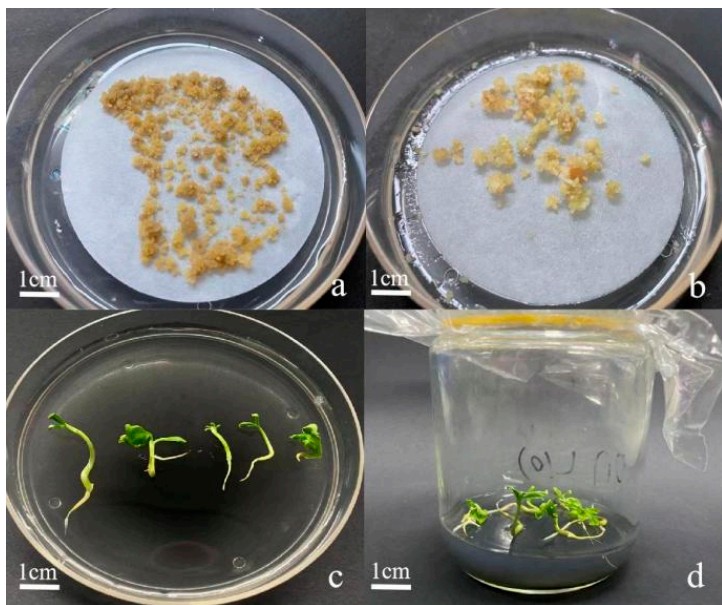

**Figure 7.** The process of recovery culture after cryopreservation of *F. mandshurica* EC by the vitrification method: (**a**) the 60th day of recovery of *F. mandshurica* EC; (**b**) callus-differentiated somatic embryo; (**c**) *F. mandshurica* buds generated from somatic embryo; (**d**) Emblings of *F. mandshurica*. bar = 1 cm.

## 4. Discussion

### 4.1. Study of Cryopreservation Methods

Cryopreservation ensures the safe and effective conservation of plant genetic resources [18]. Slow-cooling and vitrification are currently common methods of cryopreservation methods. The slow-cooling method involves dehydrating the material, placing it in a gradient cooling box and cooling it down to −80 °C at −1 °C/min. Finally, the cooled material is quickly plunged into liquid nitrogen for cryopreservation [19]. The temperature of EC of *Fraxinus excelsior* L. was reduced at a rate of −0.5/−1 °C/min [20]. However, this method is not suitable for laboratories that do not have expensive apparatus to control the temperature. This is because it can lead to a less precise rate of cooling during cryopreservation, which can affect the cooling results. A new cryopreservation technique, cryopreservation by vitrification, was developed for the cryopreservation of *F. mandshurica* EC. This method, in contrast to the published technique for cryopreservation of *F. mandshurica* EC by slow-cooling [16], does not require a temperature-controlled apparatus, does

not require a transition procedure to cryogenic cooling and is simple to operate, allowing for cryopreservation of the material in almost any laboratory. Vitrification avoids the formation of intracellular and extracellular ice crystals to a great extent, eliminates the mechanical damage caused by intracellular ice, and facilitates the entry of organs, callus and other parts to a common vitrification state. Moreover, the operation time is short, and the steps are simple. The process is unique in preserving the integrity of organs and tissue structures, and it does not need expensive cooling instruments or equipment [9,21,22]. Compared to the slow-cooling technique [16], the cell survival percentage after cryopreservation by vitrification in this paper was slightly higher than that of the slow-cooling method by 1.04%, and the regeneration percentage was slightly lower than the slow-cooling method by 2.62% (the difference was not significant). Although the percentage of regenerated plants by vitrification was slightly low, we can optimize the technical steps during future studies to improve the percentage of regenerated plants in recovery culture after cryopreservation by vitrification.

### 4.2. Effect of Pre-Culture on Vitrification Cryopreservation

In the process of cryopreservation by vitrification, steps such as pre-culture, loading, dehydration, rewarming and restoration culture, are important for the survival of materials. The vitrification method uses hyperosmotic pre-culture to reduce the content of free water in plant cells so that the cells can withstand low-temperature stress. In this study, high concentrations of sucrose were not conducive to the survival of *F. mandshurica* EC. Cryopreservation of *F. mandshurica* EC by vitrification was best suited to culture in $0.5 \text{ mol·L}^{-1}$ sucrose pre-culture medium. This finding is similar to that of cryopreservation of *Persea americana* callus [23]. In this paper, the cell survival percentage of *F. mandshurica* EC precultured with $0.5 \text{ mol·L}^{-1}$ sucrose was slightly higher than that of the slow-cooling method, and the fresh weight of EC was slightly lower than that of the slow-cooling method, but the difference between the two was not significant [16]. Loading is an indispensable step in the process of vitrification cryopreservation. Loading time is treated with cryoprotectant solution at room temperature for a certain period during cryopreservation to reduce the water content of cells and the persecution of drastic changes in osmotic pressure. The treatment time of the loading solution also affects the relative survival percentage of materials after cryopreservation. Too short or too long a time causes damage to materials and reduces the survival percentage. During the cryopreservation of *F. mandshurica* EC, the cell survival percentage decreased when the loading time was more than 60 min. This phenomenon is reflected in the callus of *Satureja spiigera* [24], the apical bud of *Smallanthus sonchifolius* [25] and the apical meristem of *Chlorophytum borivilianum* [26]. In future studies, the loading time can be reduced to improve cell survival after loading of *F. mandshurica* EC.

### 4.3. Effect of Dehydration on Vitrification Cryopreservation

During dehydration of vitrification cryopreservation, the treatment time of plant vitrification solution is the key to cell vitrification cryopreservation [27]. In this study, long dehydration time led to a decrease in the cell survival percentage of *F. mandshurica*, which is consistent with the results of the shoot tip of *Gentiana kurroo* [28] and the shoot tip of *Viola stagnina* [29]. When the dehydration time is too short, the cell dehydration is not less, and it is difficult to reach the vitrification state quickly in the process of cooling treatment. The dehydration time of *Panax ginseng* EC was 90 min [30]. Compared to the slow-cooling method [16], the dehydration time in this study was reduced by 40 min, and the cell survival percentage was increased by 8.81%, indicating that the dehydration time should be strictly controlled during the dehydration process prior to cryopreservation of *F. mandshurica* EC.

### 4.4. Study of Resuscitation Culture Conditions

The rewarming method and time are important to the cell recovery culture. The optimal rewarming time for the vitrification of *F. mandshurica* EC was 2 min in the water

bath at 40 °C. This is similar to the results of preservation of *F. mandshurica* EC by the slow-cooling method [16], the callus of *Satureja spiigera* [24], the apical meristem of *Chlorophytum borivilianum* [26] and the shoot tip of *Viola stagnina* [29]. In addition to water bath rewarming, room temperature rewarming is another alternative. The optimal rewarming method for shoot tips of *Vaccinium myrtillus* and *Allium cepa* after cryopreservation by vitrification should be performed at room temperature for 20 min [31,32]. Future research could further simplify the method of restoring healing tissue culture after cryopreservation by rewarming at room temperature to improve cell survival and regeneration plant percentage.

### 5. Conclusions

In this study, the relationships between pre-culture sucrose concentration, pre-culture time, loading time, dehydration time, rewarming method, rewarming time, callus fresh weight and cell survival percentage during cryopreservation of *F. mandshurica* EC by vitrification were determined. The technique of vitrification cryopreservation of *F. mandshurica* EC was optimized using various parameters. Specifically, the EC of *F. mandshurica* was pre-cultured in 0.5 mol·L$^{-1}$ sucrose solution at room temperature for 3 days and then cultured in liquid WPM with 2 mol·L$^{-1}$ glycerol and 0.4 mol·L$^{-1}$ sucrose for 60 min. It was then dehydrated for 50 min in PVS2 (30% glycerol + 15% DMSO + 15% glycol + 0.4 mol·L$^{-1}$ sucrose + liquid WPM) and stored in liquid nitrogen. After rewarming for 2 min in the 40 °C water bath, the preserved EC could be recovered and cultured to form intact plants.

**Author Contributions:** The authors confirm contribution to the paper as follows: L.Y., H.S., I.N.T. and A.M.N. conceived and designed the study. X.L. and Y.L. collected plant materials and prepared samples for analysis. X.L., Y.L. and X.Y. analyzed the results for experiments. X.L. and L.Y. wrote the paper. All authors have read and agreed to the published version of the manuscript.

**Funding:** The work was supported by the Fundamental Research Funds for the Central Universities (2572018BW02), the Innovation Project of State Key Laboratory of Tree Genetics and Breeding (2021B01), the National Key R&D Program of China (2017YFD0600600) and the National Natural Science Foundation of China (31400535 and 31570596).

**Data Availability Statement:** Data presented in this study are available from the corresponding author upon request.

**Acknowledgments:** We thank three anonymous reviewers and the editor for comments that improved an earlier draft of this article.

**Conflicts of Interest:** The authors declare no conflict of interest.

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
