# Peer review of "Improved Method for Cryopreservation of Embryogenic Callus of Fraxinus mandshurica Pupr. by Vitrification"

_forests, doi:10.3390/f14010028_

Round 1
Reviewer 1 Report
Introduction. Improve the introduction with cryopreservation of embryogenic calli by slow cooling techniques of species of the genus Fraxinus: Ozudogru EA, Capuana M, Kaya E, Panis B, Lambard M. Cryopreservation of Fraxinus excelsior L. embryogenic callus by one-step freezing and slow cooling techniques. Cryo Letters. 2010 Jan-Feb;31(1):63-75.
Discussion. Expand the discussion of the results of the cryopreservation of the embryogenic calli Fraxinus mandshurica with those reported for common ash (Fraxinus excelsior L.) by slow cooling. Ozudogru EA, Lambardi M. Cryotechniques for the Long-Term Conservation of Embryogenic Cultures from Woody Plants. Methods Mol Biol. 2016; 1359:537-50. doi: 10.1007/978-1-4939-3061-6_32.
Author Response
Response to Reviewer 1.
Introduction. Improve the introduction with cryopreservation of embryogenic calli by slow cooling techniques of species of the genus Fraxinus: Ozudogru EA, Capuana M, Kaya E, Panis B, Lambard M. Cryopreservation of Fraxinus excelsior L. embryogenic callus by one-step freezing and slow cooling techniques. Cryo Letters. 2010 Jan-Feb;31(1):63-75.
Discussion. Expand the discussion of the results of the cryopreservation of the embryogenic calli Fraxinus mandshurica with those reported for common ash (Fraxinus excelsior L.) by slow cooling. Ozudogru EA, Lambardi M. Cryotechniques for the Long-Term Conservation of Embryogenic Cultures from Woody Plants. Methods Mol Biol. 2016; 1359:537-50. doi: 10.1007/978-1-4939-3061-6_32.
Response.
Thank you for your professional advice. We quite agree with you. After reading these two articles carefully, we felt that the logic of the manuscript became clearer by citing them in the introduction and discussion. Preface: We have quoted the text on line 41 to make the manuscript preface logical. Discussion Section: We have cited the article at line 341 to make the manuscript discussion section more logical.
Reviewer 2 Report
Apparently, the authors did extensive work, and it seems to be interesting. However, in my opinion there is doubt whether embryogenic callus was obtained. That is, the authors are obliged to carry out histological studies of their best treatments. Histological studies give veracity to this type of work. Without this important part, the work cannot be accepted for publication.
General comments.
Apparently, the authors did extensive work, and it seems to be interesting. However, in my opinion there is doubt whether embryogenic callus was obtained. That is, the authors are obliged to carry out histological studies of their best treatments. Histological studies give veracity to this type of work. Without this important part, the work cannot be accepted for publication.
Other specific comments:
Line 83-84. Describe the acronyms 6-BA and 2,4-D.
Line 84. What does Z2(W2) refer to? Clarify in the manuscript.
Line 85. Explain why Z2 (W2) subcultured for 7-10 days were selected.
Line 88. Remove the scientific name from the header, in my opinion it is not necessary.
Line 98. Describe the acronym PVS2.
Section 2.2 should contain more headings for better understanding. Please restructure it.
Line 104, 105. Authors should describe which method they used for the relative percentage of cells. Add a subheader.
Line 113. Authors should also describe how they analyzed cell viability.
All Tables must include the units of weight. Grams, milligrams, etc.?
Lines 186-187. This must be included in all tables, even if it is repetitive. Each table must be independent.
In tables 1-6, no significant differences were really observed during the first 2 weeks, this is normal because there is a culture adaptation period. It would be desirable for the authors to carry out a mean comparison test between culture time, from day 21 to day 60.
The authors mention that they obtained embryogenic callus and show images in Figure 7, but I have doubts about it. How do the authors assure that they had embryogenic calluses when their results do not show histological sections where the formation and development of the embryos can be observed? I reviewed your previous publication (reference 15) and images related to histology are not reflected either. In fact, the callus that they report is nothing like an embryogenic callus. Authors must add histological sections related to their treatments for this study. Otherwise, they cannot ensure that they obtained embryogenic calli, which is essential for this work.
Author Response
Responses to Reviewer 2.
1) General comments.
Apparently, the authors did extensive work, and it seems to be interesting. However, in my opinion there is doubt whether embryogenic callus was obtained. That is, the authors are obliged to carry out histological studies of their best treatments. Histological studies give veracity to this type of work. Without this important part, the work cannot be accepted for publication.
Response.
First of all, thank you for your affirmation of our research, and thank you for your professional advice to us. We quite agree with you. Here is my explanation:
The F. mandshurica EC is light brown and yellow, granular. It is different from the EC of coniferous trees. We have another paper about the F. mandshurica EC as follows,
[1] Liu Y, Wei C, Wang H, Ma X, Shen HL, Yang L*. Indirect somatic embryogenesis and regeneration of Fraxinus mandshurica plants via callus tissue. J. For. Res. (2021) 32:1613–1625
https://doi.org/10.1007/s11676-020-01199-3
Figure 1 shows photographs of various periods during the differentiation process of EC of F. mandshurica without vitrification. Figure 2 shows photographs of various periods during the recovery culture of EC of F. mandshurica after cryopreservation by vitrification. Our laboratory has more than 10 years of experience in the somatic embryo culture of F. mandshurica. Both the callus of normal culture and that of recovery culture after cryopreservation could differentiate into somatic embryos, and somatic embryos could germinate into regenerated plants, which indicated that the callus did not lose embryogenesis potential after cryopreservation. This also indicates that the callus in our manuscript is EC.
The above is our explanation. We hope you can accept our explanation. Thank you very much.
Figure 1. Culture process of EC of F. mandshurica without cryopreservation by vitrification. a EC of F. mandshurica; b callus differentiated into somatic embryos after germination; c Somatic embryo seedlings of F. mandshurica. Bar=1 cm.
Figure 2. The process of recovery culture after cryopreservation of F. mandshurica EC by the vitrification method. a: cryopreservation of EC of F. mandshurica for 1d; b the 30th day of recovery of F. mandshurica EC; c the 60th day of recovery of F. mandshurica EC; d Somatic embryo differentiation from EC of F. mandshurica; e F. mandshurica buds generated from somatic embryo; f Somatic embryo seedlings of F. mandshurica.
2) Other specific comments:
- Line 83-84. Describe the acronyms 6-BA and 2,4-D.
Response. The abbreviations 6-BA and 2,4-D have been described in the manuscript. 6-BA is 6-Benzylaminopurine and 2,4-D is 2, 4-Dichlorophenoxyacetic Acid. It was highlighted in the manuscript.
- Line 84. What does Z2(W2) refer to? Clarify in the manuscript.
Response. Z2(W2) is a type of embryonic callus of Fraxinus mandshurica. An interpretation of Z2 (W2) has been added to the manuscript.
- Line 85. Explain why Z2 (W2) subcultured for 7-10 days were selected.
Response. Based on the previous research of our senior brothers and sisters, we concluded that Z2(W2) cultured at 7-10 d has the strongest differentiation ability and is most suitable for subsequent experiments.
- Line 88. Remove the scientific name from the header, in my opinion it is not necessary.
Response. The scientific name in the title has been removed.
- Line 98. Describe the acronym PVS2.
Response. The abbreviations PVS2 have been described in the manuscript. PVS2 is plant vitrification solution 2.
- Section 2.2 should contain more headings for better understanding. Please restructure it.
Response. Figures have been added to the text and the subheadings are highlighted in italics. We agree with you very much. The revised manuscript makes the method more clear.
- Line 104, 105. Authors should describe which method they used for the relative percentage of cells. Add a subheader.
Response. Thank you for your comments and we would like to explain: on line 104, the relative survival rate of cells is calculated under the heading 2.2.2.
- Line 113. Authors should also describe how they analyzed cell viability.
Response. Thank you for your comments.I'm sorry, but that was a mistake in the way we phrased it. We wanted to write "cell relative survival percentage" instead of "cell viability". Sorry again, we have revised it in the manuscript.
- All Tables must include the units of weight. Grams, milligrams, etc.?
Response. Thank you for your comments, we fully accept them. The weight of all tables is in grams and is stated in the notes under each table.
- Lines 186-187. This must be included in all tables, even if it is repetitive. Each table must be independent.
Response. We fully accept your comments. Notes have been made under all the forms in the manuscript.
- In tables 1-6, no significant differences were really observed during the first 2 weeks, this is normal because there is a culture adaptation period. It would be desirable for the authors to carry out a mean comparison test between culture time, from day 21 to day 60.
Response. Thank you for your comments.We have some explanations for your opinion: We conducted one-way ANOVA on the experimental data to determine whether each factor has a significant impact on the experimental indicators.The average comparison test you mentioned will be applied to the experiment in the following research. Thank you very much for your suggestion, which we think will be very helpful for the subsequent experiment.
The authors mention that they obtained embryogenic callus and show images in Figure 7, but I have doubts about it. How do the authors assure that they had embryogenic calluses when their results do not show histological sections where the formation and development of the embryos can be observed? I reviewed your previous publication (reference 15) and images related to histology are not reflected either. In fact, the callus that they report is nothing like an embryogenic callus. Authors must add histological sections related to their treatments for this study. Otherwise, they cannot ensure that they obtained embryogenic calli, which is essential for this work.
Response. We are in full agreement with this opinion.You mentioned this in your general comments and we have responded to your comments in the reply above.
Reviewer 3 Report
The work deals with an important issue, which is establishing a method for cryopreservation of calli of F. mandshurica by vitrification. The problem is stated correctly and solved. The authors successfully established an efficient and stable technique for the cryopreservation of F. mandshurica embryogenic callus. The manuscript is written correctly, especially the methodical part is very reliably and accurately written. The repetition of experiments in the case of this methodological work is especially important. Graphs and tables are clear. This paper is properly organized and written understandably. The manuscript gives an accurate and current representation of the scientific topic. The work includes optimization of individual steps such as effects of sucrose concentration on callus fresh weight and cell survival percentage, pre-culture time on callus fresh weight and cell survival percentage, loading time on callus fresh weight and cell survival percentage, dehydration time on callus fresh weight and cell survival percentage, rewarming methods on callus fresh weight and cell survival percentage, rewarming time on callus fresh weight and cell survival percentage.
The subject is actual, as most of the cited references come mainly from the last three years. The proper statistical methods were used.
In my opinion, the paper can be of interest to the readership of this journal, however, needs minor revisions.
The following comments should be of concern:
The aim in the abstract should be more specified the selection of the species for the study.
The first sentence of the abstract is incomprehensible.
In my opinion, numerical results should not be included in the abstract.
Why authors used basic culture conditions as it was used for cryopreservation of Anemarrhena asphodeloides? Please explain in the text.
Author Response
Responses to Reviewer 3.
- The aim in the abstract should be more specified the selection of the species for the study.
Response. Thank you for your opinion. We totally agree. We also felt that the selection of species for the study should be more clearly stated in the abstract, so we revised the first sentence of the abstract to refer to the species studied in the manuscript.
- The first sentence of the abstract is incomprehensible.
Response. Thank you for your opinion. We totally agree. We have revised the first sentence in the manuscript to clarify the meaning.
- In my opinion, numerical results should not be included in the abstract.
Response. Thank you for your comments.We fully agree with you and have deleted the numerical results in lines 26-28 of the summary.
- Why authors used basic culture conditions as it was used for cryopreservation of Anemarrhena asphodeloides? Please explain in the text.
Response. Thank you for your questions. Here is my explanation: At that time, I just considered that they are angiosperms, and I felt that the method mentioned in the article is very detailed. In the experiment, only referring to the time of vitrification cryopreservation step, the gradient was set on this basis to observe the best time suitable for the embryonic callus of Anemarrhena asphodeloides. In response to your question, we have carefully thought about it and believe that in future research, we will find species knowledge similar to my research as the reference direction.
Round 2
Reviewer 2 Report
Dear editor, the authors considered most of the observations, however, the fundamental part of the histology of somatic embryos was not incorporated, and histological studies are not observed in the publication that the authors mention. In my opinion, if they do not include at least one histological study of their best treatment, the work should not be accepted. I leave it to your consideration.
Author Response
Thank you again for your comments and we agree with you. We also thought that histological observation would be more perfect, but now because of the COVID-19 pandemic, our entire school has closed and everyone has been asked to leave school by December 15th. If we supplement the experiment, it will take 1-2 weeks to complete, so we can't supplement the experiment right now. We won't be able to start the experiment until at least March 2023, when schools reopen, and that's not guaranteed. Because of this force majeure, we can only reply you with detailed morphological photos and adequate description when you give your comments for the first time.
There was no histological study in the previous publication, because we really didn't do the histological observation, we didn't think it was necessary. Because we did obtain somatic embryos from the callus after cryopreservation by vitrification, and the somatic embryos germinated into regenerated plants, which proved that the callus after cryopreservation was embryonic, so it was the embryonic callus.
These are our explanations and we hope you will reconsider our explanations. Meanwhile, thank you again for your professional advice.